# Heterogeneity in Responding to Clinical Vignettes Depicting Sepsis Suggests That Non-Medical Data May Drive the Decision-Making Process

**DOI:** 10.3390/healthcare13202636

**Published:** 2025-10-20

**Authors:** Hossam Gad, Abdelhamed Elgazar, Krzysztof Laudanski

**Affiliations:** Department of Anesthesiology and Perioperative Care, Mayo Clinic, Rochester, MN 55902, USA; gad.hossam@mayo.edu (H.G.); elgazar.abdelhamed@mayo.edu (A.E.)

**Keywords:** clinical decision making, sepsis implementation, treatment heterogeneity, ambiguity tolerance, risk taking behavior, decision styles, clustering

## Abstract

**Background/Objectives**: Treating critically ill patients is complex and often subjective. This study investigates adherence to clinical guidelines for sepsis among different providers. Considering the strengths of the recommendations, we hypothesize that heterogeneity in the decision-making process will be low and independent of provider background and psychological makeup. **Methods**: This cohort study used two clinical vignettes of sepsis. Providers were given standardized treatment plans for 7 days, and their responses were recorded. Demographical, professional, and psychological (ambiguity tolerance, defensiveness, anxiety due to uncertainty, risk-taking behavior, decision styles, and optimism) variables were acquired. **Results**: Crystalloids were commonly used in both vignettes. Pressor engagement, especially norepinephrine, increased significantly after the third day. Providers recommended antibiotics and no provider stopped antibiotic therapy. Cluster analysis revealed no differences in therapy implementation among provider types, but some differences existed between the two vignettes. Cluster #1 was characterized by the implementation of early light bundle therapy combined with the use of pressors and a notable enhancement in therapies by the fifth day (Early Cluster). Cluster #2 (Minimalists) involved consistent engagement only in light bundle therapy throughout the treatment period. Cluster #3 (Escalation) comprised providers who rapidly escalated treatment using multiple different modalities. Cluster #3 stood out as most providers were female, non-MD, with significant ICU duties, and enhanced rational thinking. **Conclusions**: Providers differ in implementation styles of the sepsis treatment standard based on types of therapies selected not studied psychological variables.

## 1. Introduction

Treatment of critically ill patients demands skill, knowledge, and excellent decision-making abilities, all while making over 100 decisions per day [1,2]. These decisions are made under significant uncertainty, as most clinical data are imperfect and uncertain [3]. On the other hand, sepsis is an example of a situation where emphasis is placed on early implementation of several treatments even when the suspicion threshold is low [4,5,6]. Initial treatment should consist of early antibiotics implementation and fluid resuscitation. If these measures fail, pressors should be engaged. These steps are considered a gold standard in accordance with the Surviving Sepsis Campaign by the Society of Critical Care Medicine [7,8,9]. Considering that mortality increases exponentially with a treatment delay, this represents a situation where a particular combination of risk and ambiguity tolerance may interfere with providing optimal care [4]. Considering that sepsis treatment is highly standardized and the risk of error carries a significant penalty, implementation should be straightforward and thorough in cases where the level of suspicion is low. However, in some providers, the threshold for most medical decisions is not determined by simple Bayesian methods of risk-benefit ratio, but rather by their static and dynamic psychological makeup [5,10,11,12,13,14,15,16,17,18]. The literature is inconclusive on whether patient data or provider characteristics prevail, especially if the pre-test probability for the test is low.

Consequently, we embarked on testing, using clinical vignettes, how adherence to treatment varies over time and across providers. We juxtaposed a sepsis vignette with another one with less certainty, expecting increased variability in treatment. Considering the strengths of the recommendation, we hypothesize that heterogeneity in the decision-making process will be low, implementation will occur early, and it will be independent of provider background and psychological makeup [13].

## 2. Materials and Methods

### 2.1. Participants and Study Setting

This is a convenience cohort study using surveys and clinical vignettes.

This study is a part of a larger research project conducted between March 2017 and March 2018, in an academic multicenter hospital. Approved by the IRB (826741; 3 January 2018) for the University of Pennsylvania Health System, survey invitations were emailed to providers from surgery, anesthesia, and internal medicine using a pre-existing distribution list for the ICU staff. The self-administered surveys were completed via the REDCAP™ web-based survey tool. An initial recruitment email was followed by a second request sent only to those who completed the first part. We sent 1078 invitations to ICU staff and received 258 responses, although 120 questionnaires had missing variables. Only physicians and allied health professionals were included in this study, resulting in the exclusion of 55 records. The characteristics of all participants (*n* = 83) are presented in Table 1. AXIS Tool (Appraisal Tool for Cross-Sectional Studies) was utilized for assessment of potential bias [19].

### 2.2. Study Design

The participants were presented with two clinical scenarios of sepsis derived from the literature, based on real patients; however, the clinical vignettes were edited for simplification. (Appendix A Figure A1A,B) [20,21,22]. Several treatments were offered every day, which included standards for sepsis treatment by the Surviving Sepsis Campaign issued by the Society of Critical Care Medicine [7,8,9]. The combined use of antibiotics was defined as a light bundle in accordance with the existing Sepsis Campaign recommendations. Several other treatments were also presented to providers (Appendix A Figure A1A,B). The providers could choose any number of them each day. (Appendix A Figure A1A,B)

### 2.3. Psychological Variables

We utilized a few psychological tools. The Tolerance of Ambiguity Scale (TOA) examined the ability to cope with clinical uncertainty [23,24]. The Rationality/Emotional Defensiveness Scale (R/ED) assessed repression/denial [25,26]. The Anxiety Due to Uncertainty (ADU) 13-item subscale of the Physician Uncertainty Reaction measures anxiety related to ambiguity [27,28]. Propensity for risk-taking behavior was assessed using the Modified Jackson Personality Index (JPI), a six-item scale [29]. The Decision Style Survey (DSS) measured the decision-making process on two dimensions: rational and intuitive processes [30,31]. Life Orientation Test (LOT) estimates optimism [32,33]. Psychometric values of these tests are presented in the attached literature.

### 2.4. Risk of Bias Assessment

The study’s methodological quality was assessed using the AXIS tool for cross-sectional studies [19,34]. Each of the 20 domains was independently evaluated by two authors (K.L. and H.G.), and discrepancies were resolved by consensus. A summary of the AXIS assessment is provided in Appendix A Table A1.

### 2.5. Statistical Analysis

The Shapiro–Wilk W test and distribution plots were used to test the normality of distribution variables. Parametric variables will be expressed as mean ± SD and compared using t-Student. For non-parametric variables, median (M_e_) and interquartile ranges (IR) will be shown with U-Mann–Whitney statistics employed to compare such variables. We utilized *k*-means clustering with the center of initial clustering established automatically, considering we did not specify a research hypothesis for clustering analysis. The final clusters were defined after 10 iterations. A double-sided *p*-value of less than 0.05 will be considered statistically significant for all tests. Statistical analyses will be performed with Statistica 11.0 (StatSoft Inc., Tulsa, OK, USA).

## 3. Results

### 3.1. Variability in Sepsis Bundle Implementation

Our findings clearly demonstrate that the use of crystalloids was the predominant practice in both vignettes, underscoring their central role in initial treatment. While colloids and blood products were introduced later and in only limited amounts, their minimal use highlights a cautious approach. Notably, although fluid utilization showed a decline over time, this change was not statistically significance. It is also important to note that a substantial number of providers did not recommend fluid administration even at the initial stages, especially in Vignette #2 (Figure 1), raising questions about early intervention strategies. Additionally, variations in therapeutic approaches across different provider types were observed, albeit sporadically (Appendix A Figure A2). Antibiotic initiation rates varied significantly across vignettes, with less than 40% of participants starting antibiotics initially. By day 3 in vignette 2, all participants had begun full antibiotic treatment, highlighting a strong tendency toward early intervention (Figure 2). Daily adjustments to therapy were frequently made, even in less clear cases, demonstrating proactive management (Figure 2). Meanwhile, few providers discontinued treatment, indicating a potential area for optimizing antibiotic use. The implementation of protocols was consistent across provider types (Appendix A Figure A3), warrants further attention.

Pressor use increased notably after day three, with norepinephrine as the primary medication. Many providers also used dopamine and phenylephrine (Figure 3). Epinephrine was not used, and dobutamine was rarely administered (Figure 3). Approaches to pressor therapy varied slightly among providers.

Engagement with other therapies varied considerably, especially later in the treatment period when the clinical scenario for the fictional patient’s condition remained unchanged (Figure 4).

### 3.2. Clustering of Treatment

By assessing the engagement of various therapies, three clusters of recommended therapy were identified (Figure 5).

Cluster #1 featured early fluid resuscitation and pressors, with enhanced therapies by day five. Cluster #2 (Minimalists) consistently used only light bundle therapy. Cluster #3 (Escalation) providers quickly escalated treatment with multiple modalities and differed from those in clusters #1 and #2. Most providers were female, non-MD, with significant ICU duties (Table 2). Their psychological traits showed more pronounced rational thinking (Table 3).

## 4. Discussion

This study demonstrated that ICU sepsis treatment decisions displayed considerable variation, which did not correlate with providers’ demographics, professional backgrounds, or psychological characteristics. Although many practitioners followed recommended therapies, few implemented the complete sepsis bundle. Such heterogeneity was unexpected, especially given the straightforward nature of sepsis guidelines, which were anticipated to promote more uniform practices. Providers could be organized into three distinct clusters based on their implementation strategies: Cluster #1 (Early Cluster) was marked by early initiation of light bundle therapy combined with pressor use and notable intensification of treatments by day five; Cluster #2 (Minimalists) maintained consistent light bundle therapy throughout the course; and Cluster #3 (Escalation) comprised those who rapidly intensified treatment using diverse modalities. Notably, providers in Cluster #3 were predominantly female, non-physician doctors, had substantial ICU responsibilities, and exhibited enhanced critical thinking.

The evolution of guidelines since the initial edition has not resulted in significant breakthroughs [7,8,9,35,36,37]. Standard practices, such as effective infection source control with antibiotics, have remained consistent over time, along with the use of fluids and pressors to manage sepsis-related hypotension. Consistently, our findings demonstrate a low utilization of steroids, colloids, and blood products, aligning with earlier recommendations. Notably, there was a marked willingness to initiate antibiotic therapy, despite infrequent modification or discontinuation. Although current guidelines advise reassessment within 48 h, our participants typically delayed this process until the sixth or seventh day [7,38,39,40]. Achieving alignment between provider practices and established guidelines remains a challenge [37,41]. This is in stark contrast to the guidelines’ recommendations on how to treat sepsis [7]. However, one must keep in mind that we did not test how familiar these recommendations were to the providers. Also, in different countries, recommendations vary slightly, introducing a confounding variable.

Clinical guidelines are designed to deliver evidence-based recommendations for specific conditions, thereby reducing practice variability and improving mortality outcomes [37]. Previous research highlights ongoing discrepancies in ICU decision-making, particularly concerning end-of-life care, patient triage, and admission policies [42,43,44]. This variation is also evident in the management of sepsis, a well-studied condition with established guidelines aimed at reducing mortality. Implementation science suggests that this may stem from factors such as limited knowledge, disjointed handoffs, and challenges in delivering antibiotics promptly [45,46]. Although our study was theoretical and surveyed experienced critical care providers, these barriers may have still influenced their choices. While psychological traits do influence decision-making, their distribution among clinicians is generally consistent and unlikely to explain the observed variation. Existing research supports the complexity of these interactions but has yet to find definitive links.

The study’s findings should be interpreted cautiously, as it is a single-center study with unique characteristics, including a provider mix and care delivery. The study is also limited to the American healthcare providers. Generalization of these results requires validation in larger, multicenter studies. Selection bias may have occurred due to participant location and recruitment method, and the responses from non-responders remain unknown. Response rates were low, yet similar to those in comparable studies. Still, selection bias can affect our measured variables [47]. Furthermore, cluster analysis is also very limited by a relatively low response rate. These methodological issues are not uncommon, but they do caution against the generalization of our findings. We also lack insight into how participants interpreted the vignettes; conducting semi-quantitative interviews could clarify this understanding. Additionally, cognitive biases may also influence the results [48]. We did not measure knowledge of the Surviving Sepsis recommendations. The absence of correlations between the studied psychological traits and the standard implementation of sepsis is vexing, suggesting that future research should employ semi-quantitative methods to investigate the underlying decision-making processes.

The study employs established methods, including standardized psychological tests, surveys, and clinical vignettes, to assess decision-making, thereby supporting its validity [20]. The sample is broad and representative of U.S. medical professionals, with demographics comparable to those in similar research, which strengthens the internal validity.

This study clarifies how ICU providers make decisions about the sepsis bundle’s three treatment clusters. It found no association between psychological factors and decisions; however, further research is required to verify and generalize these findings.

## 5. Conclusions

Providers differ in implementation styles of the sepsis treatment standard based on types of therapies selected not studied psychological variables.

## Figures and Tables

**Figure 1 healthcare-13-02636-f001:**
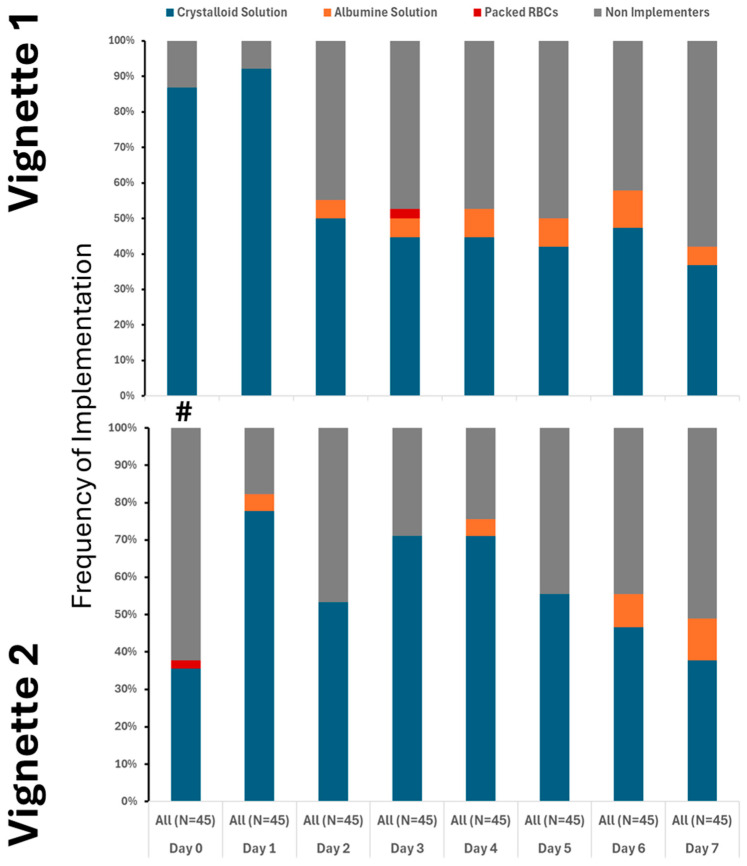
Utilization of crystalloid varied between the cases, but only on 1st day. # denotes differences in frequency of measured therapy between cases.

**Figure 2 healthcare-13-02636-f002:**
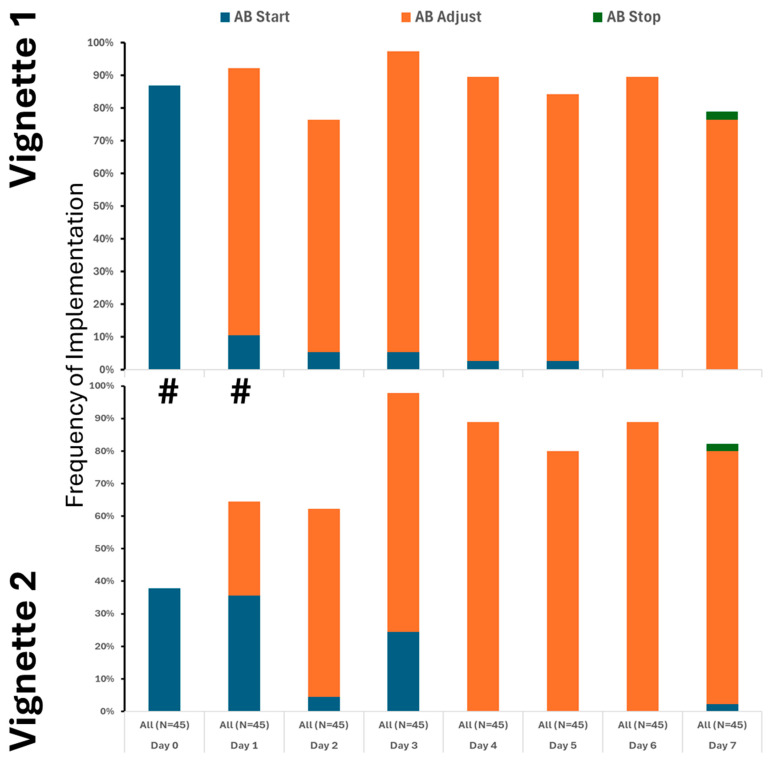
Utilization of antibiotics varied between the cases, but only on the 1st and 2nd day. # denotes differences in frequency of measured therapy between cases.

**Figure 3 healthcare-13-02636-f003:**
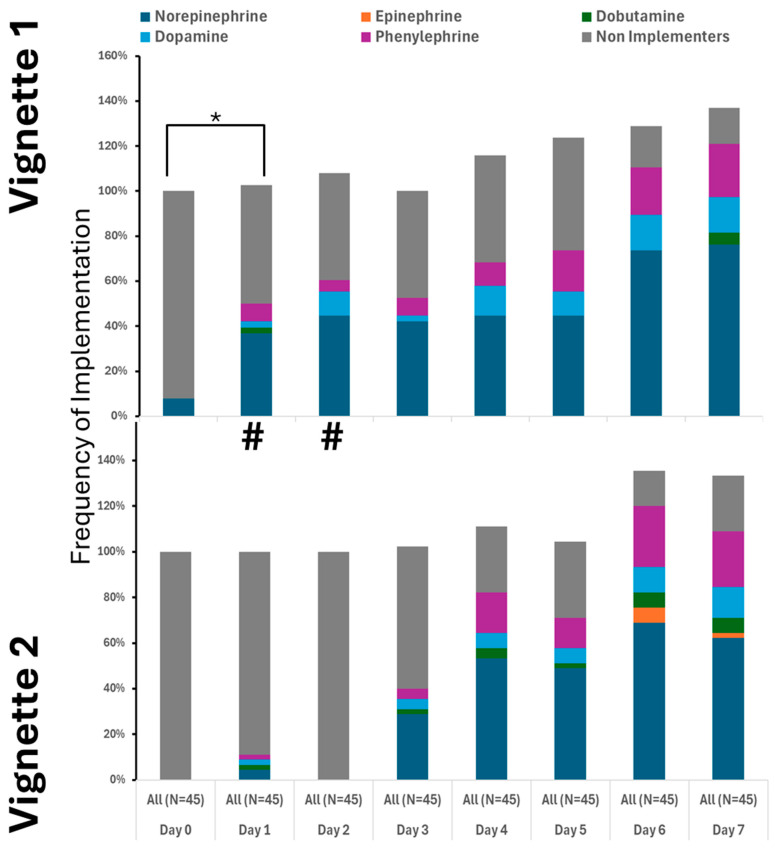
Utilization of pressors was similar between the cases. # denotes differences in frequency of measured therapy between cases. * Denotes the difference in utilized therapy to the 1st day.

**Figure 4 healthcare-13-02636-f004:**
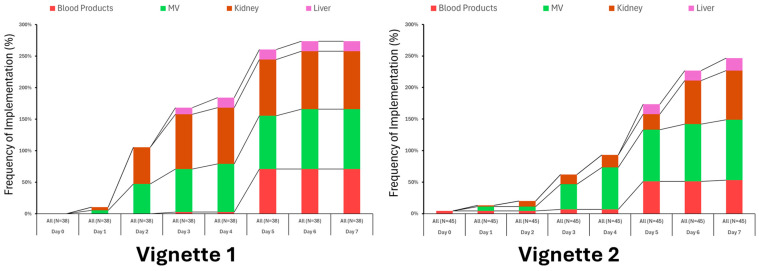
Utilization of ancillary therapies varied somewhat between vignettes but not significantly.

**Figure 5 healthcare-13-02636-f005:**
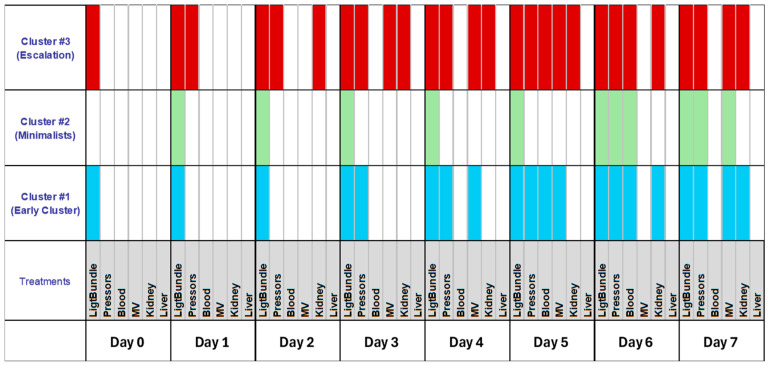
Unsupervised cluster analysis revealed three distinct clusters when treatment for sepsis was taken into consideration. The light bundle was characterized as the use of fluid resuscitation and antibiotics. MV denotes engagement of mechanical ventilation to treat evolving in a clinical scenario.

**Table 1 healthcare-13-02636-t001:** Demographic and professional characteristics of the studied group demonstrated differences, as expected from our study groups across different professionals.

	Total (*N* = 83)	Attending Physicians (*n* = 40)	Advanced Practice Provider (*n* = 43)	ANOVA *p*
**Age** *(mean ± SD)*	42.7 ± 9.97	43.5 ± 10.34	40.1 ± 9.45	ns
**Gender** *(% female)*	53.0	30.0	74.4	<0.001
**Years in Healthcare** *(mean ± SD)*	16.5 ± 10.02	16.6 ± 10.79	16.4 ± 9.36	ns
**Percent of clinical duties in ICU** *(mean ± SD)*	46.0 ± 41.04	24.3 ± 28.3	66.3 ± 40.90	<0.001
**Marital status** *(% married/co-living)*	79.5	95.0	65.1	<0.001
**People in Household** *(mean ± SD)*	2.9 ± 1.12	3.2 ± 0.99	2.7 ± 1.17	0.024

**Table 2 healthcare-13-02636-t002:** The distribution of psychological variables across the studied population revealed a significant difference in risk-taking between MDs and APPs.

Psychological Trait		Total (*N* = 83)	Attending Physicians (*n* = 40)	Advanced Practice Provider (*n* = 43)	*p*
**Tolerance for ambiguity (TOA)**		20.8 ± 5.42	20.2 ± 5.37	21.3 ± 5.49	ns
**Stress of uncertainty**		39.6 ± 8.30	39.2 ± 7.97	40.0 ± 8.67	ns
**Risk Taking**		15.9 ± 5.21	14.7 ± 5.12	17.1 ± 5.09	0.04
**Optimism**		15.3 ± 2.93	15.0 ± 2.62	15.6 ± 3.20	ns
R/ED	Defensiveness	36.4 ± 4.68	35.5 ± 4.93	37.2 ± 4.34	ns
	Optimistic denial	43.5 ± 9.39	43.3 ± 9.98	43.7 ± 8.93	ns
**Decision-making Scale**	Rational decision making	4.2 ± 0.48	4.2 ± 0.49	4.3 ± 0.48	ns
	Intuitive decision making	2.7 ± 0.57	2.7 ± 0.57	2.8 ± 0.57	ns

**Table 3 healthcare-13-02636-t003:** Basic demographic and professional characteristics of the studied group demonstrated differences as expected across different clusters. The distribution of psychological variables across the various clusters revealed a significant difference in rational decision-making.

		Total (*N* = 83)	Cluster #1 (*n* = 30)	Cluster #2 (*n* = 21)	Cluster #3 (*n* = 32)	ANOVA *p*
**Age** ** *(mean ± SD)* **		42.7 ± 9.97	41.0 ± 11.08	42.9 ± 9.44	41.7 ± 9.44	ns
**Gender** *(% female)*		53.0	43.3	38.1	**71.9**	0.023
**Years in Healthcare** *(mean ± SD)*		16.5 ± 10.02	15.0 ± 10.89	17.1 ± 9.26	17.5 ± 9.77	ns
**Percent of clinical duties in ICU** *(mean ± SD)*		46.0 ± 41.04	39.2 ± 37.22	32.7 ± 38.99	**61.1 ± 42.17**	0.023
**Marital status** *(% married/co-living)*		79.5	83.3	76.2	78.1	ns
**Healthcare Role** *(% MD)*		48.2	60	61.9	**28.1**	0.015
**People in Household** *(mean ± SD)*		2.9 ± 1.12	2.9 ± 1.06	3.14 ± 1.11	2.78 ± 1.12	ns
**Tolerance for Ambiguity (TOA)**		20.8 ± 5.42	23.6 ± 5.41	24.4 ± 3.43	16.2 ± 4.09	ns
**Stress of uncertainty**		39.6 ± 8.30	52.2 ± 4.84	41.3 ± 4.28	33.8 ± 5.57	ns
**Risk Taking**		15.9 ± 5.21	13.2 ± 4.56	15.2 ± 3.62	17.0 ± 4.88	ns
**Optimism**		15.3 ± 2.93	13.7 ± 3.61	14.9 ± 2.01	14.1 ± 2.91	ns
**R/ED**	**Defensiveness**	36.4 ± 4.68	37.8 ± 4.57	34.5 ± 4.46	36.5 ± 4.57	ns
	**Optimistic denial**	43.5 ± 9.39	35.5 ± 6.63	41.4 ± 4.19	35.7 ± 5.93	ns
**Decision-making Scale**	**Rational decision making**	4.2 ± 0.48	4.3 ± 0.42	4.2 ± 0.40	**4.4 ± 0.47**	0.04
	**Intuitive decision making**	2.7 ± 0.57	2.6 ± 0.57	2.9 ± 0.53	2.6 ± 0.55	ns

## Data Availability

The data presented in this study are available on request from the corresponding author. The data are not publicly available due to privacy and ethical restrictions.

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
