# Peer review of "Heterogeneity in Responding to Clinical Vignettes Depicting Sepsis Suggests That Non-Medical Data May Drive the Decision-Making Process"

_healthcare, 2025, doi:10.3390/healthcare13202636_

Round 1
Reviewer 1 Report
Comments and Suggestions for Authors
Introduction
Well written introduction, no comments.
Methods
No comment.
Results
Final sample is small, with subgroup analyses (clusters, provider type) possibly underpowered. Stability of cluster analysis with n=83 is questionable.
Sadly you cannot change that, but only address it in discussion.
Discussion
Out of 1078 invitations you received 258 responses, 120 had missing data, and ended up with 83 participants.
So out of 1,078 invited, only 83 were actually analyzed – that’s 7.7% of the original sample.
This raises some concerns.
The combination of low response rate, high missing data, and exclusions raises concerns about representativeness and possible selection bias. The authors should discuss how these limitations may affect the validity and generalizability of their findings.
The authors appropriately note heterogeneity in decision-making, but conclusions about “not driven by psychological characteristics” may be premature given sample size and attrition.
You should state that more cautious interpretation is warranted, and that these findings are hypothesis-generating.
Author Response
Comments 1: Final sample is small, with subgroup analyses (clusters, provider type) possibly underpowered. Stability of cluster analysis with n=83 is questionable.
Sadly you cannot change that, but only address it in discussion.
- Response 1: We agree what there are inherited limitations of the study. We added a suggestion to discussion
Comments 2: Out of 1078 invitations you received 258 responses, 120 had missing data, and ended up with 83 participants.So out of 1,078 invited, only 83 were actually analyzed – that’s 7.7% of the original sample.
This raises some concerns.
- Response 2: We added this point ot the discussion. Actually, such a poor response rates are common for this kind of study. Nevertheless, there is concern for bias.
Comments 3: The combination of low response rate, high missing data, and exclusions raises concerns about representativeness and possible selection bias. The authors should discuss how these limitations may affect the validity and generalizability of their findings.
- Response 3: We added limitations to the discussion
Comments 4: The authors appropriately note heterogeneity in decision-making, but conclusions about “not driven by psychological characteristics” may be premature given sample size and attrition. You should state that more cautious interpretation is warranted, and that these findings are hypothesis-generating.
- Response 4: We introduced modifications as suggested.
Reviewer 2 Report
Comments and Suggestions for Authors
This is a very interesting single center survey based study with small number of subjects (n<100) based on survey for which clinical vignettes have been used.
I have a few recommendations:
Explain bundle therapy as not all readers would be well-versed with the term. What is light bundle therapy?
I am unsure if the statement that the treatment of sepsis is subjective - please provide more details and references on the reasons that it is subjective. Some of it is explained in the discussion but this line is inappropriate for the abstract. Treatment is not subjective but there is "subjective variation" in treatment when clinicians have difficulties with follow guidelines.
Please add in the methods that this was a simulated scenario with clinical vignettes without real patients.
Please expand on what is MV in figure 4
What cluster followed the guidelines? That is unclear from the text
Please assess the internal validity with Newcastle-Ottawa or appropriate scale meant for such survey based study. This will objectively tell the readers the quality of the study in question. You can include this as a table in the manuscript.
Important to note that vignette based study only correlate moderately to real life. Add references on how much do the vignette based study correlate to real life
Measurement of proper care. I do not see how the proper care was defined? This is ambiguous, especially for the uncertain cases
Overall, I think this a underpowered survey based study and real life applicability remains a question. Although the topic is worth exploring further
Author Response
Comments 1: Explain bundle therapy as not all readers would be well-versed with the term. What is light bundle therapy?
- Response 1: It is the concomitant use of fluid resuscitation and antibiotics in accordance with the Surviving Sepsis Campaign. We described this in the text (materials and methods) and in several figure legends
Comments 2: I am unsure if the statement that the treatment of sepsis is subjective. Please provide more details and references on the reasons why it is subjective. Some of it is explained in the discussion but this line is inappropriate for the abstract. Treatment is not subjective but there is "subjective variation" in treatment when clinicians have difficulties with follow guidelines.
- Response 2: We agree with the suggestion. Wording was improved.
Comments 3: Please add in the methods that this was a simulated scenario with clinical vignettes without real patients.
- Response 3: These were fictional clinical scenarios based on literature and editorialized content.
Comments 4: Please expand on what is MV in figure 4
- Response 4: Added to the legend
Comments 5: What cluster followed the guidelines? That is unclear from the text
- Response 5: This section now provides a more detailed description of the clustering technique.
Comments 6: Please assess the internal validity with Newcastle-Ottawa or an appropriate scale meant for such a survey-based study. This will objectively tell the readers the quality of the study in question. You can include this as a table in the manuscript.
- Response 6: The Newcastle–Ottawa Scale (NOS) is a tool used to assess the quality of non-randomized studies included in a systematic review and/or meta-analyses. Considering the pilot nature of this study, we do not believe that another tool needs to be addressed to support our discussion in the last chapter – this is very preliminary data. Additionally, the non-randomized tools were not standardized for clinical vignettes, raising concerns about potential errors related to using a tool designed for different types of studies in this methodology.
Comments 7: It is important to note that vignette-based studies only moderately correlate with real life. Add references on how much vignette-based studies correlate with real life.
- Response 7: I believe this is a highly subjective assessment. We base the vignettes on case reports published in NEJM. A group of experts altered them to maintain their core, make them credible, and ensure clinical relevance. A separate panel manages these tasks. We also made minor adjustments to the materials and methods section. References were included.
Comments 8: Measurement of proper care. I do not see how the proper care was defined? This is ambiguous, especially for the uncertain cases
- Response 8: The Surviving Sepsis Campaign, developed by the Society of Critical Care Medicine, was utilized. The initial description provided was insufficiently detailed. An appropriate introduction to the standard of care for sepsis has now been incorporated into the text.
Comments 9: Overall, I think this is an underpowered survey-based study, and real-life applicability remains a question. Although the topic is worth exploring further
- Response 9: We agree with the reviewer. This is a small study with several biases and will need validation. The results are puzzling, and we address these points in more detail in the discussion.
Reviewer 3 Report
Comments and Suggestions for Authors
- Page 1, line 33-34, the following statement is not always correct “These decisions are made under significant uncertainty, as most clinical data are imperfect and uncertain.”
- Page 5, line 117-18, states “There was slight variation between providers (data not shown) in the approach to antibiotic therapy.” If it is antibiotic therapy, then this statement should come in the previous paragraph. If it is pressor therapy, kindly correct it.
- Page 7, line 132 mentions about “early light bundle therapy.” Kindly define it or add reference about the same.
- Page 7, line 136-37, states “Cluster #3 stood out compared to clusters #1 and #2.”Kindly make it a complete sentence, stood out with respect to what?
- Page 9, line 159-60 states “Standard practices such as effective infection source control using fluids and pressors have remained consistent over time.” This statement is wrong. Infection control involves taking sample for culture and early initiation of antibiotics, preferably within the first hour. Fluid and pressors are for management of hypotension.
- Expand ISBRED.
- While the article states “Also, cognitive biases may play role in affecting the results” it is also a fact that proper teaching and training forms the basis of effective implementation of any guideline. Stated so, presence or absence of any data regarding prior training of the participants in surviving sepsis guideline will add to the merit or limitation of the study
Author Response
|
|
|
Comments 1: Page 1, line 33-34, the following statement is not always correct “These decisions are made under significant uncertainty, as most clinical data are imperfect and uncertain.” |
|
· Response 1: We change the language slightly. We do believe, and data support this, that a lot of sepsis treatment is done when probabilities are vexing. |
|
Comments 2: Page 5, line 117-18, states “There was slight variation between providers (data not shown) in the approach to antibiotic therapy.” If it is antibiotic therapy, then this statement should come in the previous paragraph. If it is pressor therapy, kindly correct it. |
|
· Response 2: We appreciate catching this problem. It has been remedied. |
|
Comments 3: Page 7, line 132 mentions about “early light bundle therapy.” Kindly define it or add reference about the same. |
|
· Response 3: Defined |
|
Comments 4: Page 7, line 136-37, states “Cluster #3 stood out compared to clusters #1 and #2.”Kindly make it a complete sentence, stood out with respect to what? |
|
· Response 4: We rewrote this paragraph |
|
Comments 5: Page 9, line 159-60 states “Standard practices such as effective infection source control using fluids and pressors have remained consistent over time.” This statement is wrong. Infection control involves taking sample for culture and early initiation of antibiotics, preferably within the first hour. Fluid and pressors are for management of hypotension. |
|
· Response 5: I think we are both wrong. The current recommendation (cited) suggests giving antibiotics early, even if this means a sample for culture will not be obtained. What we tried to say is that pressors and fluid resuscitation are common recommendations in guidelines across several years. They are used to treat sepsis-related hypotension. We modified the text to be more clear. |
|
Comments 6: Expand ISBRED. |
|
· Response 6:IBS was a typo. The name of the scale has been corrected |
|
Comments 7: While the article states “Also, cognitive biases may play role in affecting the results” it is also a fact that proper teaching and training forms the basis of effective implementation of any guideline. Stated so, presence or absence of any data regarding prior training of the participants in surviving sepsis guideline will add to the merit or limitation of the study |
|
· Response 7: Excellent points. We implemented them in the discussion. |
Round 2
Reviewer 2 Report
Comments and Suggestions for Authors
Authors have adequately revised the article and addressed the concerns. I still believe a risk of bias assessment tool should be included. If they do not want to use Newcastle tool, they should include any other validated tool for survey based studies. There are several available and it should be included to enhance the transparency of the study
Author Response
We utilized AXIOS tool to assess the risk of bias